# Mean Platelet Volume-to-Albumin Ratio as a Predictor of Mortality in Patients with Febrile Neutropenia: An Observational Study

**DOI:** 10.3390/medicina61040601

**Published:** 2025-03-26

**Authors:** Jelena Dimitrijević, Marina Čalamać, Ognjen Đurmez, Marko Stojanović

**Affiliations:** 1Institute for Oncology and Radiology of Serbia, 11000 Belgrade, Serbia; drjelenadimitrijevic@gmail.com (J.D.); marina.calmac@ncrc.ac.rs (M.Č.); ognjen.djurmez@ncrc.ac.rs (O.Đ.); 2Department of Pharmacology, Clinical Pharmacology and Toxicology, Faculty of Medicine, University of Belgrade, 11000 Belgrade, Serbia

**Keywords:** febrile neutropenia, MPV-to-albumin ratio, MASCC risk index, prediction of mortality

## Abstract

*Background and Objectives*: Febrile neutropenia (FN) is a life-threatening complication in cancer patients, necessitating early risk stratification for optimal management. While the Multinational Association for Supportive Care in Cancer (MASCC) risk index is commonly used for mortality prediction, novel biomarkers that integrate inflammatory and nutritional status may improve prognostic accuracy. This study evaluates the mean platelet volume (MPV)-to-albumin (ALB) ratio as a predictor of mortality in patients with febrile neutropenia. *Materials and Methods*: A prospective study was conducted on patients diagnosed with febrile neutropenia. The MPV/ALB ratio was calculated at admission, and its predictive value for mortality was assessed using receiver operating characteristic (ROC) curve analysis. The results were compared with the MASCC risk index for predicting mortality both during the FN episode and within 28 days. Statistical analysis included sensitivity, specificity, and negative predictive value assessments. *Results*: The MPV/ALB ratio was significantly lower in non-survivors compared to survivors. A threshold of 0.245 effectively stratified patients at high risk, demonstrating strong specificity and high negative predictive value. The MPV/ALB ratio outperformed the MASCC risk index in predicting mortality during febrile neutropenia, whereas the MASCC index showed slightly better performance for 28-day mortality prediction. *Conclusions*: The MPV/ALB ratio is a promising biomarker for mortality prediction in febrile neutropenia, offering a simple and accessible tool for early risk assessment. Its superior performance in predicting mortality during FN highlights its potential clinical utility. Further prospective studies are needed to validate these findings and explore the integration of the MPV/ALB ratio into existing risk assessment models.

## 1. Introduction

Febrile neutropenia (FN) is a frequent complication following standard treatments in cancer patients [1]. Its occurrence varies, impacting approximately 10% to 50% of those with solid tumors and up to 80% of patients with hematologic cancers [1,2]. FN is recognized as a medical emergency due to the serious complications and negative outcomes that it brings, affecting patient survival considerably. Studies have shown a higher rate of in-hospital mortality associated with FN [3]. Although FN significantly raises mortality risk soon after it appears, this impact generally decreases over time [3]. Prolonged FN episodes are also linked to worse survival rates, underscoring the need for timely intervention in managing symptoms [4]. FN increases vulnerability to various infections, including bloodstream (bacteremia and septicemia), respiratory, gastrointestinal, urinary tract, and skin and soft tissue infections [5,6,7]. Furthermore, FN can lead to mucositis, potentially allowing bacterial entry into the bloodstream and leading to secondary infections [5,6]. Infections acquired in the hospital setting during FN tend to be more resistant to treatment, contributing further to the risk of fatal outcomes [8].

Due to the significant influence of FN on cancer patient treatment and prognosis, anticipating its complications and outcomes is highly important. Many studies have explored the potential role of biomarkers, especially those linked to inflammation, in forecasting FN-related outcomes and complications [7,9,10,11,12]. With inflammation and malnutrition being common issues among cancer patients and their harmful effects on prognosis [13], there is significant interest in examining whether biomarkers tied to inflammation and malnutrition can forecast FN outcomes and complications [12]. One potential biomarker that could serve this purpose is the MPV/ALB (mean platelet volume-to-albumin) ratio. The MPV/ALB ratio may offer valuable insights into the inflammatory and nutritional status of patients with FN. Specifically, MPV is often elevated during systemic inflammation [14], reflecting platelet activation and the body’s inflammatory response, while serum albumin levels typically decrease during acute inflammation or stress and indicate underlying nutritional status [15,16,17,18]. This combination suggests that the MPV/ALB ratio could provide a composite measure of systemic stress in FN patients. Although the MPV/ALB ratio is present, it is neither widely used in clinical practice nor extensively researched. Nevertheless, its ability to capture both inflammation and nutritional status makes it promising for identifying high-risk patients early, potentially guiding more personalized and intensive management strategies.

Building on these considerations, this study aimed to evaluate the potential of the MPV/ALB ratio as an early predictor of mortality during FN and its ability to forecast 28-day mortality outcomes. Additionally, we compared its predictive performance to the MASCC score, which is currently regarded as the gold standard for this purpose.

## 2. Materials and Methods

The study adhered to the principles outlined in the STROBE (Strengthening the Reporting of Observational Studies in Epidemiology) guidelines for observational research.

### 2.1. Study Design

This prospective observational study took place at the Institute for Oncology and Radiology of Serbia, a tertiary-level university hospital located in Belgrade, which also functions as the National Cancer Research Center. The research was carried out over a 20-month period, starting in May 2022 and ending in December 2023.

A well-defined protocol governed biomarker measurement. Blood samples for MPV and albumin were obtained immediately upon hospital admission as part of the FN diagnostic workup. The analysis of these biomarkers was conducted promptly; ensuring results were accessible within an hour of confirming the diagnosis.

### 2.2. Study Population and Sample Size Calculation

The sample size estimation indicated that 185 patients were needed for the study, accounting for a 5% margin of error and ensuring an 80% confidence level. The study included both male and female patients aged 18 and above, all diagnosed with either solid or hematologic malignancies, who had experienced at least one episode of FN during the research period. Patients who developed FN while receiving combined chemotherapy and radiotherapy, or radiotherapy alone, were excluded from participation.

### 2.3. Statistical Analysis

The data were first gathered using Microsoft Excel 2016 and later imported into GraphPad Prism 10 Software (GraphPad Software, San Diego, CA, USA) for detailed statistical evaluation. The normality of the data distribution was checked using the D’Agostino and Pearson omnibus normality test. Based on the distribution pattern, the results were reported as either mean with standard deviation (SD) or median with interquartile range (IQR).

We assessed the ability of MPV/albumin ratio to predict complications and outcomes related to FN by generating receiver operating characteristic (ROC) curves. These ROC curves were compared to those created for the MASCC risk index. To identify the optimal cut-off point for each ROC curve, we utilized Youden’s J statistic, which allowed us to calculate sensitivity, specificity, positive predictive value, negative predictive value, and the likelihood ratio.

The cut-off values from the MPV/Albumin ratio ROC curves for mortality and 28-day mortality were used to reclassify patients into two groups. After reclassification, a Kaplan–Meier curve was created to compare the survival of patients from these two groups during FN and 28-day survival. To estimate the difference in survival during FN and 28-day survival, the Kaplan–Meier log-rank test was used. A *p*-value less than 0.05 was considered to be statistically significant.

### 2.4. Ethical Approval

The performed study was in line with the institutional and national research committees’ ethical standards. The study followed the 1964 Helsinki Declaration and its later amendments. The Institute for Oncology and Radiology of Serbia Ethical Review Board initially approved this study protocol under the number no. 1184-01, afterwards the Research Ethical Committee of the Medical Faculty, University of Belgrade, granted the approval of the study with document number no. 1322/VII-15.

Informed consent was obtained from all participating patients prior to their enrollment in the study. Written approval was obtained from each patient after they were provided with detailed information regarding the study objectives and procedures. Patients were assured that their participation was voluntary and that they had the right to withdraw at any time.

To ensure confidentiality and privacy of participants, all study data were analyzed anonymously. Personally identifiable information was removed or replaced with unique identifiers before data were analyzed. Only authorized personnel had access to the coded data, and any identifying information was securely stored separately from the analyzed dataset.

## 3. Results

### 3.1. Study Population and Demographic Characteristics

In this study, we analyzed data collected from 185 individuals diagnosed with FN, as documented by the Institute for Oncology and Radiology of Serbia. The study population had a median age of 57, with an age range of 18 to 86. The male-to-female ratio among participants was 41.30% to 59.24%, with 76 male subjects and 109 female subjects. The majority of participants had solid tumors (80%; *n* = 148). Anthropometric parameters indicated a median weight of 65.50 kg and a median BMI of 22.12 kg/m^2^. Laboratory measurements showed a median absolute neutrophil count of 0.1 × 10^9^/L and a median white blood cell count of 1.1 × 10^9^/L. Ninety-one participants (49.46%) had a MASCC score ≥ 21, while 93 participants (50.54%) had a score lower than 21. Among the study population, 16 participants (8.70%) developed bacteremia, 9 (4.89%) developed septicemia, 15 participants (8.15%) died during FN, and 33 participants (17.93%) died within the first 28 days. The median length of hospital stay was 7 days. The key demographic characteristics are summarized in Figure 1.

### 3.2. Predictive Accuracy of MPV/ALB Ratio

The ROC curve for the MPV/ALB ratio demonstrated its accuracy in predicting mortality during FN (AUC = 0.79; 95% CI: 0.66–0.91, *p* = 0.0002) and 28-day mortality (AUC = 0.70; 95% CI: 0.59–0.80, *p* = 0.0005) (Figure 2). Similarly, the MASCC score proved effective in predicting mortality during FN (AUC = 0.75; 95% CI: 0.64–0.88, *p* = 0.001) and 28-day mortality (AUC = 0.75; 95% CI: 0.66–0.84, *p* < 0.0001). Interestingly, the MPV/ALB ratio exhibited a higher AUC than the MASCC risk index for predicting mortality during FN. Conversely, the MASCC risk index score showed a slightly higher AUC value for predicting 28-day mortality when compared to the MPV/ALB ratio.

The MPV/ALB ratio demonstrated varying predictive performance for mortality during FN and for 28-day mortality (Table 1). It showed moderate sensitivity for both predicting mortality during FN and 28-day mortality. Additionally, the MPV/ALB ratio exhibited moderate specificity for predicting 28-day mortality, while its specificity for predicting mortality during FN was strong. The positive predictive value of the MPV/ALB ratio for both mortality outcomes was low, whereas the negative predictive values were high. Furthermore, the MPV/ALB ratio presented moderate likelihood ratios, indicating moderate evidence that it increases the likelihood of dying during FN or within the first 28 days.

### 3.3. Kaplan–Meier Survival Analysis for MPV/ALB Ratio

After calculating the MPV/ALB cut-off values from the ROC curve that predicted overall mortality during FN (cut-off = 0.245) and 28-day mortality (cut-off = 0.245), we classified patients into two groups, low and high MPV/ALB values, and created a Kaplan–Meier curve to investigate the differences in survival between these groups. After dividing patients into two groups, the Kaplan–Meier curve revealed that a lower MPV/ALB ratio is associated with a higher mortality rate during FN (Figure 3A). Similarly, the Kaplan–Meier curve indicated that a low MPV/ALB ratio is linked to increased mortality rates within the 28 days following the development of FN (Figure 3B). Kaplan–Meier statistical indicators are presented in Table 2.

## 4. Discussion

This study highlights the significant prognostic value of the MPV-to-albumin ratio as a reliable predictor of mortality risk among patients with febrile neutropenia, both during the acute phase and in the 28-day period following diagnosis. The MPV/ALB ratio outperformed the MASCC risk index in predicting mortality during febrile neutropenia, while the MASCC risk index showed a slight advantage over the MPV/ALB ratio for predicting 28-day mortality. The MPV/ALB ratio offers moderate predictive utility for both mortality during FN and 28-day mortality, with strong specificity for mortality during FN and high negative predictive values across both outcomes. Although its positive predictive value is low, the MPV/ALB ratio moderately increases the likelihood of predicting mortality risk in these patients. A lower MPV/ALB ratio (below 0.245) has a higher risk of mortality both during the FN episode and within the following 28 days. The Kaplan–Meier survival curves underscore the prognostic value of the MPV/ALB ratio in identifying patients at greater mortality risk in these critical periods.

The outcomes of FN can vary depending on the patient’s individual characteristics and underlying condition [1]. Key factors influencing the severity of FN include the patient’s age, the presence of comorbidities, and the type and stage of cancer. As a consequence of the FN, different complications, such as infectious diseases sepsis and pneumonia, can develop, or mortality can occur. The complications that arise due to FN can significantly impact the initial treatment approach, often necessitating reductions in cytostatic doses or delays in treatment cycles. These adjustments may compromise the effectiveness of cancer therapy and affect overall survival rates. More importantly, in addition to complications, FN itself increases the risk of mortality, particularly during the episode and, to a slightly lesser degree, in the early period following FN resolution [19]. FN is associated with a high mortality rate, ranging from 10% to 30% [20,21,22,23]. The mortality risk linked to FN, especially during the acute phase, coupled with the potential impact of comorbidities, underscores the critical need for timely risk prediction in these patients. In cancer supportive care, ongoing efforts to enhance the accuracy of risk prediction are vital, given the complexity and severity of FN-related complications and outcomes.

A frequently investigated approach in FN research focuses on using biomarkers to predict patient outcomes. Among these, MPV reflects inflammation, while albumin is notable for its ability to indicate both inflammatory and nutritional status, both of which are critical factors in FN. Previously, these biomarkers were studied individually across various clinical settings. However, the MPV/ALB ratio has not been extensively investigated as a predictor of mortality in critically ill patients. This is largely because initial MPV values are not generally considered reliable prognostic markers of mortality in such populations [24]. Moreover, it is often recommended to consider the third-day MPV values when assessing the predictive potential of this marker in critically ill patients [24], which is not meaningful in critical conditions. Given the duration and severity of FN, MPV values measured on the third day would not be ideal. The optimal time for MPV measurement in this case is at the time of diagnosis.

In previous studies, albumin has been established as a valuable predictor of mortality among hospitalized patients [25], with lower levels correlating with an increased risk of mortality [26]. Additionally, in our previous study, we investigated and compared albumin levels to MASCC score values for mortality prediction in FN patients [12]. The results showed that albumin performed comparably to the MASCC risk score, with both having the same AUC value (AUC = 0.75; *p* = 0.001), though albumin did not surpass the MASCC score as the standard reference. For this reason, we sought to enhance the predictive power of albumin by incorporating an additional inflammatory marker (MPV). This approach led to improved mortality prediction in FN patients and even surpassed the MASCC score’s predictive ability in this context.

In contrast to its ability to predict mortality during FN, the MPV/ALB ratio did not outperform albumin alone in predicting 28-day mortality in FN patients. Our previous research demonstrated that albumin was a significant predictor of 28-day mortality in FN patients (AUC = 0.70; *p* = 0.0002), though it fell short of the predictive power of the MASCC risk index score [12]. Incorporating MPV into a ratio with albumin yielded statistically significant results, but the AUC value (0.70) remained identical to that of albumin alone and did not surpass the predictive ability of the MASCC risk index score (AUC = 0.75).

The AUC values of the MPV/ALB ratio for predicting mortality (0.79) and 28-day mortality (0.70) are generally classified as fair [27]. For a prognostic factor to be considered significant, its AUC value must exceed 0.5, while an AUC of 0.8 or higher is typically regarded as optimal [27]. However, an AUC below 0.8 may still be deemed optimal for both diagnostic and prognostic purposes if it surpasses the AUC value of the established gold standard, which in this case is the MASCC score [27].

The sensitivity and specificity of the MPV/ALB ratio for predicting mortality during FN suggest that, due to its relatively low sensitivity and moderately high specificity, this ratio is more effective at identifying individuals unlikely to die than those who will. The low sensitivity of this predictive tool contributed to its limited positive predictive value (PPV) [28]. This reduced sensitivity is likely a consequence of the low incidence of mortality observed in our study. Specifically, 14 out of 185 participants (8.15%) died during FN, which is significantly lower than the mortality rates reported in the literature, where incidences can reach up to 30% [28]. In addition to its lower AUC values, the performance of the MPV/ALB ratio for predicting 28-day mortality was also unsatisfactory compared to its performance in predicting mortality during FN. The low sensitivity and specificity observed when evaluating the MPV/ALB ratio for 28-day mortality cumulatively contributed to a low PPV [28,29,30], as both factors play a role in reducing the proportion of true positives relative to the total number of positive test results.

After determining the cut-off values for both mortality during FN and 28-day mortality from respective ROC curves, these thresholds were used to reclassify patients into low- and high-risk groups. Kaplan–Meier mortality curves were then created based on this classification. Both curves showed that patients in the high-risk group had significantly higher mortality rates compared to those in the low-risk group. Notably, the Kaplan–Meier curve for mortality during FN revealed that almost all deaths occurred within the high-risk group, a pattern not observed for the 28-day mortality curve. These findings further highlight that the MPV/ALB ratio is a more accurate predictor of mortality during FN than of 28-day mortality.

The findings of this study underscore the clinical significance of the MPV/ALB ratio as a valuable prognostic tool for mortality risk assessment in patients with febrile neutropenia. Given its ability to outperform the MASCC risk index in predicting mortality during the FN episode, this biomarker ratio may serve as an adjunct or alternative for early risk stratification, particularly in settings where comprehensive scoring systems may not be immediately available. Its strong specificity and high negative predictive value suggest that the MPV/ALB ratio could aid in identifying lower-risk patients, potentially guiding clinical decision-making regarding hospitalization, intensive monitoring, or treatment escalation. While the MASCC risk index remains the gold standard for FN prognosis, the MPV/ALB ratio offers a readily accessible and cost-effective biomarker-based approach that warrants further validation in larger, multicentric studies to explore its full clinical utility.

This study has several limitations that should be addressed. Firstly, as a single-center study, the findings may be difficult to generalize to broader populations. Secondly, most participants had solid tumors, which limits the applicability of the results to patients with hematologic malignancies. Thirdly, the potential impact of varied antibiotic regimens and granulocyte colony-stimulating factor administration on patient outcomes cannot be discounted, as these factors may have influenced the observed results. Additionally, while our tests demonstrated utility, improvements in sensitivity and specificity could further enhance the validity and reliability of our findings.

The complexity of malignancy and chemotherapy intensity significantly impact FN severity and mortality, but were not considered in our study. Our study primarily focused on the MPV/ALB ratio as a predictor of mortality and did not include a detailed analysis of malignancy type or chemotherapy intensity as independent variables. While these factors are crucial in FN prognosis, we believe our use of the MASCC score, which incorporates several clinical risk factors related to malignancy and treatment intensity, provides a comprehensive risk assessment. Future studies could explore how malignancy complexity and chemotherapy intensity specifically interact with MPV/ALB in predicting mortality.

Infection duration and antibiotic treatment length can influence FN outcomes. However, in clinical practice, antibiotic duration is determined by established FN treatment guidelines, which include clear criteria for discontinuation. Therefore, we do not consider antibiotic duration itself to be a primary determinant of patient outcomes; rather, the severity and course of FN itself play a crucial role. Prolonged treatment is usually a consequence of more complicated FN cases, which inherently carry a higher risk of adverse outcomes.

We recognize that some FN complications, like acute kidney injury (AKI), respiratory failure, and heart failure, can significantly contribute to increased mortality. However, our study primarily focused on evaluating the MPV/ALB ratio as a predictor of mortality, without specifically assessing the role of individual organ failures in diagnostic and treatment accuracy. While these complications undoubtedly impact clinical outcomes, our study was not designed to analyze their direct effects on diagnostic markers or treatment decisions. Instead, we used the MASCC score, which incorporates various clinical parameters relevant to FN severity, to provide a comprehensive risk stratification. Future research could explore how organ dysfunction (AKI, respiratory, and heart failure) interacts with MPV/ALB in predicting mortality and guiding treatment adjustments. We appreciate this valuable perspective and will consider it in future studies.

We need to acknowledge that while potential correlation between the quick Sequential Organ Failure Assessment (qSOFA) score and the MPV/ALB ratio could provide additional insights into organ dysfunction in FN, our study primarily focused on comparing MPV/ALB with the MASCC score. The MASCC score was chosen because it is specifically designed for FN, whereas qSOFA is a more generalized sepsis-related scoring system.

## 5. Conclusions

This study demonstrates the potential of the MPV-to-albumin ratio as a valuable biomarker for predicting mortality risk in patients with febrile neutropenia, both during the acute phase and within the first 28 days following diagnosis. The findings suggest that a lower MPV/ALB ratio (below 0.245) is associated with an increased risk of mortality in these critical periods. Notably, the MPV/ALB ratio outperformed the MASCC risk index in predicting mortality during febrile neutropenia, while the MASCC risk index exhibited a slight advantage for predicting 28-day mortality. The strong specificity and high negative predictive values of the MPV/ALB ratio further highlight its clinical relevance.

## Figures and Tables

**Figure 1 medicina-61-00601-f001:**
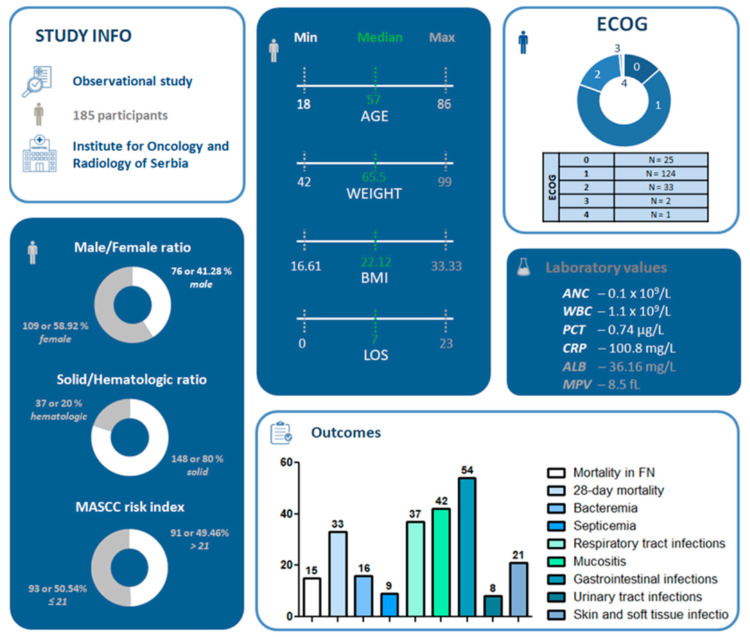
Study infographic presenting demographic characteristics and key outcomes. Laboratory values are expressed as median. ALB—albumin, ANC—absolute neutrophil count, BMI—body mass index, CRP—C-reactive protein, ECOG—eastern cooperative oncology group, FN—febrile neutropenia, LOS—length of stay, MASCC—multinational association of supportive care in cancer, MPV—mean platelet volume, PCT—procalcitonin, WBC—white blood cell.

**Figure 2 medicina-61-00601-f002:**
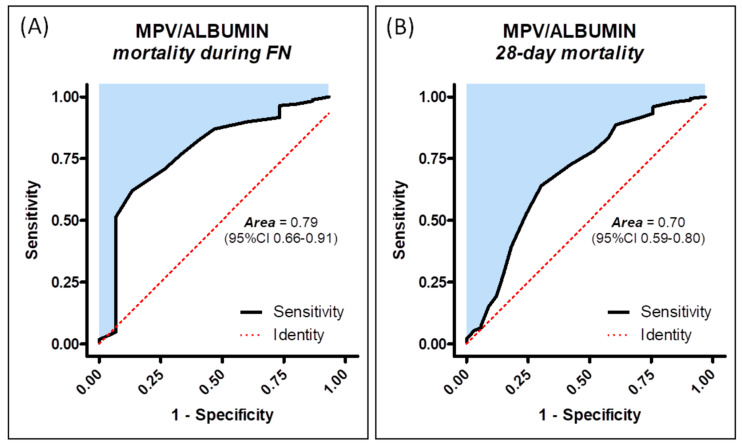
Receiver operating characteristic curves for prediction of mortality during FN (**A**) and 28-day mortality (**B**).

**Figure 3 medicina-61-00601-f003:**
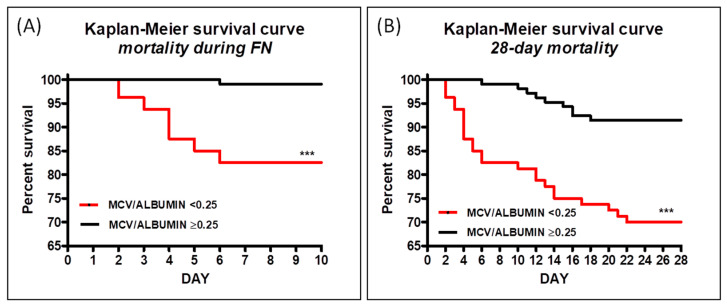
Kaplan–Meier survival curve for mortality during FN (**A**) and 28-day mortality (**B**). *** *p*< 0.001.

**Table 1 medicina-61-00601-t001:** Prognostic performance of MPV/ALB ratio in predicting mortality during FN, and 28-day mortality.

MPV/ALB	Sensitivity	Specificity	PPV	NPV	Likelihood Ratio
Mortality in FN	0.62 (0.54–0.69)	0.87 (0.60–0.98)	0.29 (0.1–0.75)	0.96 (0.94–0.97)	4.77
28-day mortality	0.64 (0.56–0.72)	0.69 (0.51–0.84)	0.31 (0.20–0.50)	0.90 (0.84–0.93)	2.06

MPV—mean platelet volume; NPV—negative predictive value; PPV—Positive predictive value.

**Table 2 medicina-61-00601-t002:** Kaplan–Meier statistical indicators.

MPV/ALBUMIN	Chi Square	Hazard Ratio (95%CI)	*p* Value
Mortality during FN	16.94	8.79 (3.12–24.73)	<0.0001
28-day mortality	15.22	4.04 (2.00–8.14)	<0.0001

## Data Availability

The datasets generated and analyzed during this study are available from the corresponding author upon reasonable request.

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
