# Peer review of "Mean Platelet Volume-to-Albumin Ratio as a Predictor of Mortality in Patients with Febrile Neutropenia: An Observational Study"

_medicina, 2025, doi:10.3390/medicina61040601_

Round 1
Reviewer 1 Report
Comments and Suggestions for Authors
This is a nicely written manuscript, which is based on the investigating the role of Mean Platelet Volume to Albumin ratio as a predictor of mortality in patients with Febrile Neutropenia.
This study included data collection from 185 individuals who were diagnosed with Febrile Neutropenia.
The manuscript is of interest to clinical researchers and internal medicine in aspects of investigating the importance of the promising biomarker for mortality prediction in febrile neutropenia, offering a simple and accessible tool for early risk assessment.
In my opinion, this study approach provides much useful information about the role of exploring the integration of the MPV/ALB ratio into existing risk assessment models.
I would recommend this study to be accepted after a few minor corrections and provide brief explanations:
- There are challenges with the complexity of the malignancy related to the intensity of chemotherapy. So, how did authors utilize this approach in this study?
- Longer infections in Febrile Neutropenia may affect by the antibiotic treatment duration and affect the outcome of the treatment, so how do authors report this issue?
- Do authors consider the consequence of Febrile Neutropenia symptoms in acute kidney injury, respiratory and heart failure in high mortality and how it affects the diagnostic and treatment accuracy in this study?
- Authors should focus more on the significance and usefulness of this study.
Thanks
Author Response
1. There are challenges with the complexity of the malignancy related to the intensity of chemotherapy. So, how did authors utilize this approach in this study?
Thank you for your insightful question. We acknowledge that the complexity of malignancy and chemotherapy intensity significantly impacts febrile neutropenia (FN) severity and mortality. However, our study primarily focused on the Mean Platelet Volume-to-Albumin (MPV/ALB) ratio as a predictor of mortality and did not include a detailed analysis of malignancy type or chemotherapy intensity as independent variables.
While these factors are crucial in FN prognosis, we believe our use of the MASCC score, which incorporates several clinical risk factors related to malignancy and treatment intensity, provides a comprehensive risk assessment. Future studies could explore how malignancy complexity and chemotherapy intensity specifically interact with MPV/ALB in predicting mortality.
2. Longer infections in Febrile Neutropenia may affect by the antibiotic treatment duration and affect the outcome of the treatment, so how do authors report this issue?
We appreciate your thoughtful input. We agree that infection duration and antibiotic treatment length can influence FN outcomes. However, in clinical practice, antibiotic duration is determined by established FN treatment guidelines, which include clear criteria for discontinuation. Therefore, we do not consider antibiotic duration itself to be a primary determinant of patient outcomes; rather, the severity and course of FN itself play a crucial role. Prolonged treatment is usually a consequence of more complicated FN cases, which inherently carry a higher risk of adverse outcomes.
3. Do authors consider the consequence of Febrile Neutropenia symptoms in acute kidney injury, respiratory and heart failure in high mortality and how it affects the diagnostic and treatment accuracy in this study?
We recognize that acute kidney injury (AKI), respiratory failure, and heart failure are critical complications in febrile neutropenia (FN) that can significantly contribute to increased mortality. However, our study primarily focused on evaluating the Mean Platelet Volume-to-Albumin (MPV/ALB) ratio as a predictor of mortality, without specifically assessing the role of individual organ failures in diagnostic and treatment accuracy.
While these complications undoubtedly impact clinical outcomes, our study was not designed to analyze their direct effects on diagnostic markers or treatment decisions. Instead, we used the MASCC score, which incorporates various clinical parameters relevant to FN severity, to provide a comprehensive risk stratification.
Future research could explore how organ dysfunction (AKI, respiratory, and heart failure) interacts with MPV/ALB in predicting mortality and guiding treatment adjustments. We appreciate this valuable perspective and will consider it in future studies.
4. Authors should focus more on the significance and usefulness of this study.
We have now added a dedicated paragraph at the end of the Discussion section to explicitly highlight the significance and clinical usefulness of our study. This addition emphasizes the MPV/ALB ratio's potential role in improving early risk stratification for febrile neutropenia, particularly in settings where rapid decision-making is critical. We also discuss its advantages over the MASCC risk index in predicting mortality during FN and its potential as a cost-effective, biomarker-based tool for guiding clinical management. We believe this revision strengthens the overall impact of our findings and provides a clearer perspective on the study’s practical implications.
Reviewer 2 Report
Comments and Suggestions for Authors
The study conducted by Dimitrijević et al. evaluates the mean platelet volume (MPV)-to-albumin (ALB) ratio as a predictor of mortality in patients with febrile neutropenia.The MPV/ALB ratio was calculated at admission, and its predictive value for mortality was assessed using receiver operating characteristic (ROC) curve analysis. The results were compared with the MASCC risk index for mortality prediction during the FN episode and within 28 days. Statistical analysis included sensitivity, specificity, and negative predictive value assessments.
Comments:
1. Fig 1 could use a better color scheme, the data would be more readable
2. The Kaplan Meier survival curves are well defined
Author Response
1. Fig 1 could use a better color scheme, the data would be more readable
Thank you for your constructive suggestion regarding Figure 1. We agree that improving the color scheme could enhance readability. In response, we have adjusted the color palette to ensure better contrast and clarity, making the data more visually accessible. We appreciate your feedback, as it has helped us refine the presentation of our results.
2. The Kaplan Meier survival curves are well defined
We sincerely appreciate your positive feedback on the Kaplan-Meier survival curves. We aimed to present the data in a clear and interpretable manner, and we are glad that this aspect of the study was well received.
Reviewer 3 Report
Comments and Suggestions for Authors
The manuscript is well written and easy to follow. I have some minor suggestions
- Authors should avoid using abbreviations in the title
- Although figure 1 depicts the details about study population, it will be better to present it in the form of tables, adding more details such as grade, stage, other comorbidities, therapeutic regimen, CBC parameters of the patients (RBC distribution width, delta neutrophil index)etc.
- If available, it will be a good idea to correlate quick Sequential Organ Failure Assessment (qSOFA) score with MPN/ALB biomarkers
Author Response
1. Authors should avoid using abbreviations in the title
We appreciate the reviewer’s suggestion. The abbreviations have been removed, and the title has been updated to: “Mean Platelet Volume-to-Albumin Ratio as a Predictor of Mortality in Patients with Febrile Neutropenia: An Observational Study.”
2. Although figure 1 depicts the details about study population, it will be better to present it in the form of tables, adding more details such as grade, stage, other comorbidities, therapeutic regimen, CBC parameters of the patients (RBC distribution width, delta neutrophil index)etc.
The main study information has been presented in an infographic format to provide a fast and clear visual overview of the key findings. However, to ensure a more detailed understanding of the study population characteristics, we have included additional information within the text of the manuscript. If needed, we can further elaborate on specific parameters as suggested.
3. If available, it will be a good idea to correlate quick Sequential Organ Failure Assessment (qSOFA) score with MPN/ALB biomarkers
Thank you for your valuable suggestion regarding the potential correlation between the quick Sequential Organ Failure Assessment (qSOFA) score and the MPV/ALB ratio. While we agree that qSOFA could provide additional insights into organ dysfunction in febrile neutropenia (FN), our study primarily focused on comparing MPV/ALB with the MASCC score, which is a well-established tool for risk stratification in FN patients. The MASCC score was chosen because it is specifically designed for febrile neutropenia, whereas qSOFA is a more generalized sepsis-related scoring system. Additionally, we did not originally plan to collect all qSOFA components (e.g., respiratory rate, mental status), making retrospective calculation infeasible and potentially introducing bias due to missing data. Our findings highlight the potential role of MPV/ALB as a prognostic biomarker in FN, particularly in relation to the MASCC score. Future studies could explore the integration of qSOFA and MPV/ALB, but this was beyond the scope of our current analysis.